# Synergistic effects of *Clostridium butyricum* and *Akkermansia muciniphila*-derived postbiotics ameliorate DSS-induced colitis and associated tumorigenesis through immunomodulation and microbiota regulation in mice

Dengxiong Hua,[1,2,3] Qin Yang,[1,2,3] Xuexue Zhou,[1,2,3] Daoyan Wu,[1,2,3] Yingqian Kang,[1,2,3] Lei Tang,[4] Boyan Li,[5,6] Zhengrong Zhang,[1,2,3] Xinxin Wang,[1,2,3] Wei Hong,[1,2,3] Zhenghong Chen,[1,2,3] Guzhen Cui[1,2,3]

**ABSTRACT** Inflammatory bowel disease (IBD) is a major precursor to colorectal cancer (CRC). Our previous study demonstrated that combined administration of the probiotics *Clostridium butyricum* (CB) and *Akkermansia muciniphila* (AKK) significantly alleviated IBD and CRC symptoms in mice. Increasing evidence suggests that probiotic metabolites (postbiotics) offer significant advantages in disease prevention and treatment without the stability and safety concerns associated with live bacterial therapies. To further explore the therapeutic potential of CB- and AKK-fermented metabolites against IBD and CRC, we established a DSS-induced IBD model and DSS/AOM-induced CRC orthotopic models in mice and evaluated the effects of CB and AKK metabolites on alleviating IBD and CRC. The results revealed that the fermented metabolites of CB and AKK (designated as SupCB and SupAKK, respectively) exhibited significant synergistic effects. Mixed fermented metabolites (designated as SupCBAKK) outperformed individual metabolites, significantly alleviating IBD and CRC symptoms by modulating immune responses, repairing the mucosal barrier, and ameliorating gut dysbiosis. Notably, SupCBAKK synergized with the immune checkpoint inhibitor anti-PD-L1 (aPD-L1), enhancing tumor sensitivity to immunotherapy and amplifying antitumor immune responses. These findings underscore the potential of SupCBAKK as a novel postbiotic formulation for mitigating IBD and CRC progression and offer innovative strategies for developing CB- and AKK-based therapeutic interventions.

**IMPORTANCE** This study highlights the therapeutic potential of SupCBAKK, a novel postbiotic formulation derived from the combined fermentation metabolites of CB and AKK, IBD, and colitis-associated colorectal cancer through the modulation of gut microbiota and immunometabolism.

**KEYWORDS** inflammatory bowel disease (IBD), colorectal cancer (CRC), *Clostridium butyricum*, *Akkermansia muciniphila*, postbiotics

Inflammatory bowel disease (IBD) and colitis-associated colorectal cancer (CRC) are complex diseases driven by chronic intestinal inflammation and microbial dysbiosis (1). Studies have shown that patients with IBD have a significantly increased risk of developing CRC compared with healthy individuals (1). Currently, biological agents such as anti-TNF-α monoclonal antibodies (e.g., infliximab) and immune checkpoint inhibitors (e.g., aPD-L1) have improved clinical outcomes for IBD/CRC patients to some extent; however, two major challenges remain (2, 3): (i) Approximately 30%–40% of IBD/CRC patients do not respond to existing therapies (4), and (ii) the "cold" tumor

**Peer Reviewers** Yan Wang, University of Maryland Baltimore, Baltimore, Maryland, USA; Bidisha Barat, The University of Chicago, Chicago, Illinois, USA

Address correspondence to Guzhen Cui, cuiguzhen@gmc.edu.cn, Zhenghong Chen, chenzhenghong@gmc.edu.cn, or Wei Hong, hongwei@gmc.edu.cn.

Dengxiong Hua, Qin Yang, Xuexue Zhou, and Daoyan Wu contributed equally to this article. Co-first authors are listed in order of their relative contributions

The authors declare no conflict of interest.

See the funding table on p. 17.

microenvironment of microsatellite stable (MSS) CRC reduces sensitivity to immunotherapy, leading to a significant decrease in the efficacy of anti-PD-1/PD-L1 treatment (5). Recent studies have found that probiotics exhibit significant efficacy in alleviating the progression of IBD/CRC, remodeling the gut microbiota, modulating the immune microenvironment, and transforming the "cold-hot" tumor state (6). Thus, exploring new probiotic modulation strategies to improve IBD/CRC progression has become an important direction of current research.

*Clostridium butyricum* (CB) and *Akkermansia muciniphila* (AKK) are natural probiotics present in the intestines of healthy humans or animals and play significant roles in protecting the intestinal barrier, modulating gut microbiota, and enhancing host immunity (2, 7, 8). In our previous studies, we found that live CB and AKK exhibited remarkable synergistic effects, effectively alleviating weight loss, colon shortening, and disease activity index (DAI) in mice while also modulating gut microbiota composition and suppressing colonic inflammatory responses (9). Moreover, the combination of CB and AKK (CBAKK) significantly enhanced the sensitivity of CRC mice to the immune checkpoint inhibitor aPD-L1, improving antitumor efficacy and survival rates. These findings provide new experimental evidence and strategies for the prevention and treatment of IBD and CRC (9).

However, live bacterial therapy faces numerous challenges in clinical translation, such as the need for comprehensive research on strain cultivation, survival rate, colonization efficiency, and safety (10, 11). Postbiotics are a class of biologically active substances produced by probiotics through fermentation and are rich in beneficial components, such as short-chain fatty acids (SCFAs), amino acids, enzymes, and extracellular polysaccharides. They exhibit various functions, including modulation of host immunity, improvement of the gut microbiota, repair of the intestinal barrier, and promotion of digestion and absorption (12–15). Compared with live bacteria, probiotic fermentation products demonstrate higher stability in inflammatory environments, are not influenced by host microbiota colonization rates, and avoid potential infection risks (16). These characteristics make postbiotics a promising alternative to live bacterial therapy, with significant research value and application potential in the food and pharmaceutical industries.

Previous research on CB fermentation metabolites has primarily emphasized short-chain fatty acids (e.g., acetate, propionate) (17, 18), which significantly inhibit pathogen activity, invasiveness, and biofilm formation, while also modulating intestinal inflammation and microbial balance (2, 19, 20). In contrast, AKK fermentation products are rich in organic acids and lipid metabolites, which hold considerable potential for pathogen control and metabolic disease intervention (21–24). Importantly, AKK produces a unique outer membrane protein, Amuc_1100, which enhances tight junctions, promotes epithelial repair, and alleviates DSS-induced barrier damage and endotoxemia (8, 24, 25). Interestingly, both live and pasteurized AKK demonstrate comparable therapeutic efficacy in IBD treatment, underscoring the functional relevance of bacterial metabolites (8). Thus, both CB and AKK cells and their secreted products play essential roles in host immune regulation and microecological balance. Although previous studies have focused on the functional roles of metabolic products from single bacteria (either CB or AKK) in disease treatment, there have been no reports on the use of dual-bacterial mixed fermentation to prepare a composite postbiotic.

The individual fermentation products of CB and AKK contain a variety of bioactive components, including extracellular vesicles, secreted proteins such as Amuc_1100 from AKK, and key small molecules such as butyrate, acetate, and lactate (26–28). Nevertheless, studies on postbiotics have primarily emphasized the overall functionality of composite products, which are often more effective than single metabolites. Building on the significant therapeutic effects of live CB and AKK in treating IBD and CRC observed in our previous studies, we further isolated the fermentation supernatants of CB, AKK, and their combination (termed SupCB, SupAKK, and SupCBAKK) to evaluate their biological roles. Our findings demonstrated that the fermentation products of CB

and AKK effectively alleviated the progression of IBD and CRC in mice by modulating host immunity and reshaping the gut microbiota. Moreover, these metabolites enhanced the responsiveness of CRC mice to the immune checkpoint inhibitor aPD-L1, thereby improving antitumor efficacy. Notably, the dual-strain fermentation product SupCBAKK exhibited significantly superior therapeutic outcomes compared with single-strain fermentation products. Collectively, this study provides a novel avenue for developing "live bacteria-free" therapeutic strategies against IBD and CRC and promotes the clinical translation of postbiotics-based combination immunotherapy.

## RESULTS

### The combined fermentation metabolites of CB and AKK (SupCBAKK) demonstrated superior efficacy in alleviating DSS-induced colitis in mice compared with single-strain fermentation products

To evaluate the functional effects of CB and AKK fermentation metabolites in alleviating DSS-induced colitis in mice, we administered C57BL/6 mice with culture medium (DSS control group), CB fermentation supernatant (SupCB group), AKK fermentation supernatant (SupAKK group), or CB/AKK combined fermentation supernatant (SupCBAKK group) via oral gavage. Mice treated with saline alone served as the healthy control group (Fig. 1A). We found that both SupCB and SupAKK metabolites restored body weight and colon length in mice. However, the therapeutic efficacy of the SupCBAKK combined treatment group was significantly superior to that of the single treatment group (Fig. 1B through E). Furthermore, the results demonstrated that SupCB, SupAKK, and SupCBAKK combination group all alleviated DSS-induced colonic tissue damage, reduced inflammatory cell infiltration, and restored crypts and mucosal layers. Notably, the SupCBAKK combination group exhibited the most significant alleviation effect ($P = 0.0012$, Fig. 1F and G). Therefore, we compared the expression levels of Zonula Occludens-1 (ZO-1) protein in the intestines of mice across groups using Immunohistochemistry (IHC). The results revealed that the SupCBAKK combination group had the highest ZO-1 protein expression level compared with the SupCB and SupAKK groups ($P = 0.0237$, Fig. 1H and I). In summary, these findings suggest that the combined metabolites of the two strains were superior to the single-strain metabolites in effectively alleviating DSS-induced colitis in mice.

### SupCBAKK activates host immunity and effectively alleviates inflammatory responses in colitis mice

To analyze the effects of CB and AKK fermentation metabolites on pro-inflammatory cytokines in colitis mice, we measured the expression levels of IL-1β, IL-6, and TNF-α in the colonic tissues of mice from each group using ELISA. The results showed that, compared with the disease control group (DSS), the expression levels of IL-1β, IL-6, and TNF-α in the colonic tissues of mice treated with SupCBAKK were reduced by 67.24% ($P < 0.0008$), 77.7% ($P < 0.0202$), and 67.0% ($P < 0.0059$), respectively (Fig. 2A through C), indicating that SupCBAKK can mitigate inflammatory responses in colitis mice.

P65 and TLR4 are key regulators of the NF-kB signaling pathway and play critical roles in modulating inflammatory progression. Therefore, we examined the expression levels of these two factors in colonic tissues by western blotting (WB). The results revealed that, after SupCBAKK treatment, the expression levels of P65 and TLR4 in colonic tissues were reduced by 59.49% and 43.51%, respectively (Fig. 2D and E). In summary, these findings demonstrate that SupCBAKK effectively activated host immunity, significantly reduced the expression of pro-inflammatory cytokines in colonic tissues, and suppressed inflammatory responses, thereby effectively alleviating DSS-induced intestinal barrier dysfunction in mice (Fig. 2F).

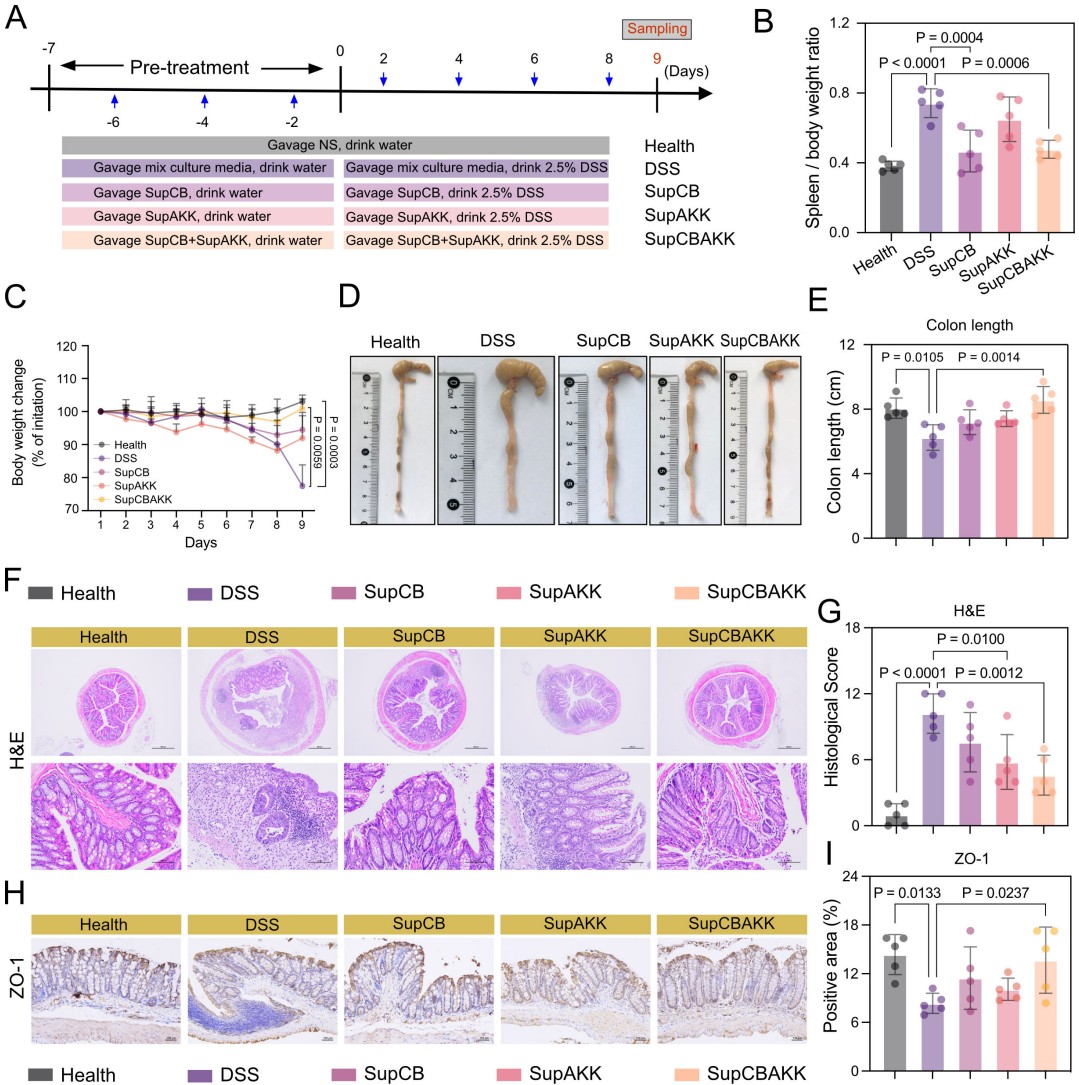

**FIG 1** SupCBAKK ameliorates DSS-induced colitis in mice. (A) Schematic representation of the experimental design for establishing and treating the DSS-induced colitis mouse model. (B) Spleen-to-body weight ratio in experimental mice. (C) Temporal changes in body weight throughout the experimental period. (D) Representative images of mouse colon anatomy and (E) measurement of colon length. (F) Representative photomicrographs of H&E-stained colon sections, and (G) histopathological scoring (scale bars: 500 µm and 100 µm). (I) Immunohistochemical staining of colon tissues and (H) quantitative assessment of ZO-1-positive areas (scale bar: 100 µm). Data are presented as means ± SD (*n* = 5). Statistical analysis was performed using one-way ANOVA followed by Tukey's multiple comparison test. Significant differences compared with the DSS group are denoted as *$P < 0.05$, **$P < 0.01$, ***$P < 0.001$, ****$P < 0.0001$.

## SupCBAKK modulated gut microbiota in colitis mice by increasing the abundance of beneficial bacteria and reducing the abundance of harmful bacteria

To evaluate the effects of CB and AKK fermentation metabolites on the gut microbiota of colitis mice, we collected fecal samples from each group and analyzed the changes in the gut microbiota before and after metabolite treatment using 16S rDNA amplicon sequencing. Principal coordinate analysis (PCoA) revealed that the gut microbiota of DSS-induced colitis mice was significantly altered compared with that of healthy mice. However, the gut microbiota composition was significantly restored in the SupCBAKK intervention group, with a more pronounced improvement compared with the single-strain metabolite interventions (Fig. 3A). The microbial dysbiosis index (MDI) analysis was consistent with the PCoA results, showing that the gut microbiota in the SupCBAKK intervention group was most similar to that in the healthy group (Fig. 3B). Additionally,

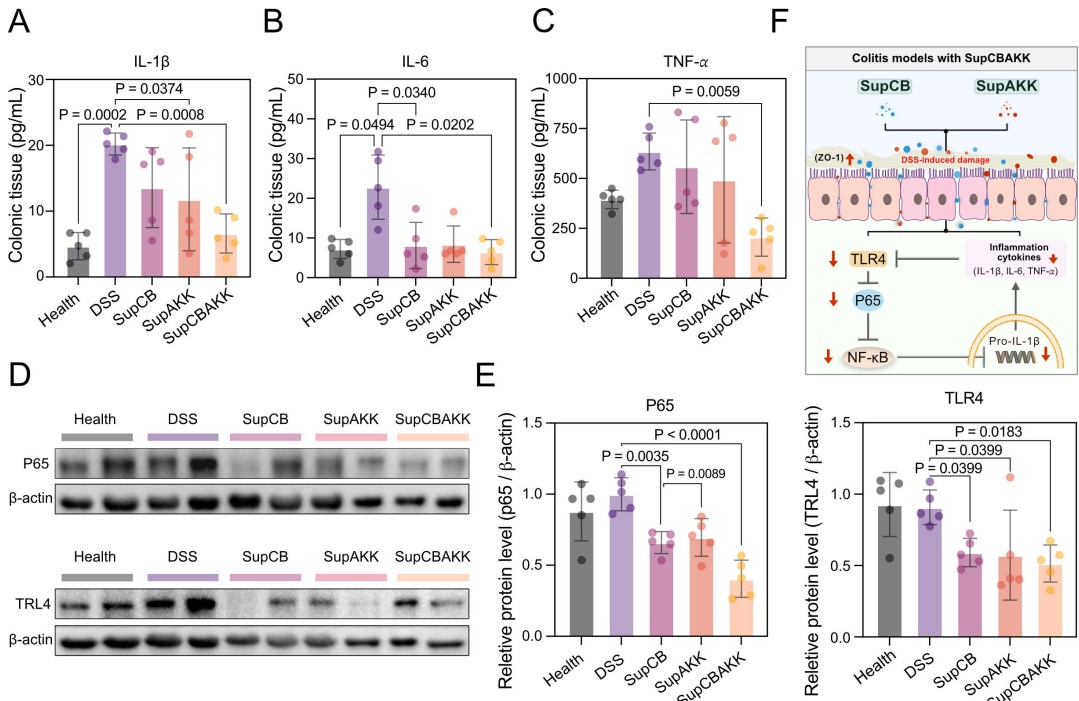

**FIG 2** SupCBAKK attenuates intestinal inflammatory responses in DSS-induced colitis mice. (A–C) ELISA analysis of pro-inflammatory cytokines IL-1β (A), IL-6 (B), and TNF-α (C) in colonic tissues. (D) Representative western blot images demonstrating the protein expression levels of p65 and TLR4. (E) Quantitative analysis revealed that SupCBAKK treatment significantly downregulated the expression of key mediators p65 and TLR4, which are associated with NF-κB signaling pathway activation. (F) Schematic illustration of the proposed mechanism by which SupCBAKK ameliorates intestinal inflammation in DSS-induced colitis mice. Data are presented as means ± SD ($n = 5$). Statistical analysis was performed using one-way ANOVA followed by Tukey's multiple comparison test. Significant differences compared with the DSS group are denoted as *$P < 0.05$, **$P < 0.01$, ***$P < 0.001$, ****$P < 0.0001$.

the Veen results showed that there were 101 common species across all groups, whereas the unique operational taxonomic units (OTUs) in the healthy group, disease group, SupCB group, SupAKK group, and SupCBAKK group were 11, 18, 13, 6, and 5, respectively (Fig. 3C).

To further analyze the characteristics of different species among the groups, we compared the abundance and levels of microbial communities and individual species between groups using community bar plot analysis. We found that the SupCBAKK intervention group exhibited significant bidirectional regulatory features in gut microbiota reconstruction (Fig. 3D and E). On one hand, SupCBAKK intervention significantly increased the positive enrichment of beneficial genera with intestinal protective effects, such as *Lactobacillus*, *norank_o_Clostridia_UCG-014*, *Akkermansia*, and *Desulfovibrio* (Fig. 3F through J). On the other hand, the SupCBAKK intervention effectively suppressed the overgrowth of harmful bacteria in colitis mice, including *Escherichia-Shigella* and *Sutterella* (Fig. 3K and L). In summary, microbiome analysis revealed that after SupCBAKK intervention, the abundance of beneficial bacteria in the mouse intestine significantly increased, whereas the abundance of harmful bacteria significantly decreased. The gut microbiota composition in these mice became more similar to that of healthy mice, indicating that SupCBAKK ameliorated DSS-induced microbiota dysbiosis in colitis mice.

## SupCBAKK modulates the gut microbiome-immune axis to alleviate colitis in mice

To further elucidate the microbiota regulatory characteristics of SupCBAKK intervention, we employed the LEfSe algorithm to analyze the cross-taxonomic-level microbiome differences between the SupCBAKK group and the DSS disease control group. Multidimensional comparisons revealed 20 significantly different taxonomic units between the

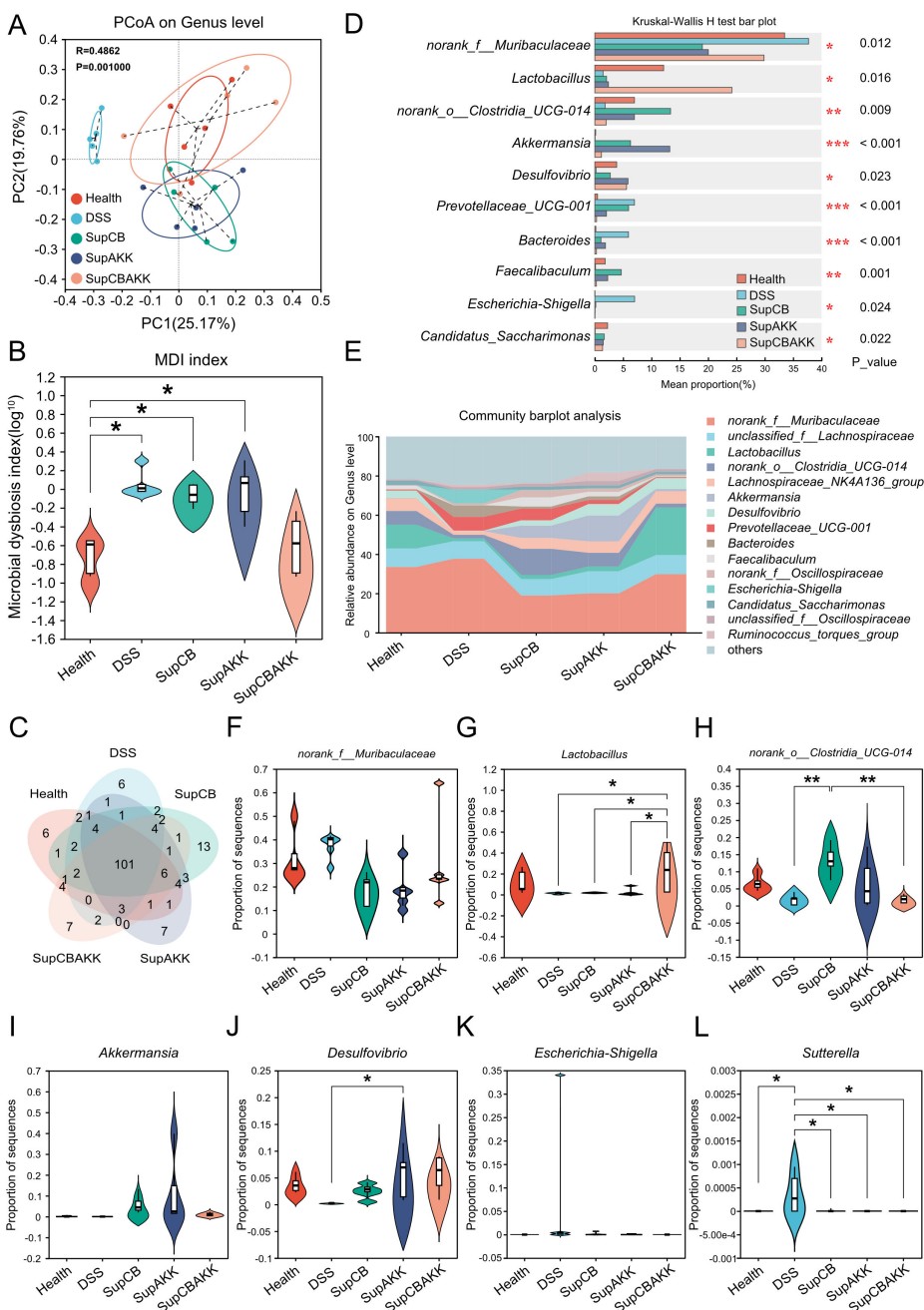

**FIG 3** SupCBAKK restores gut microbiota diversity in DSS-induced colitis mice. (A) Differential assessment of the relative abundance of fecal microbiota among five groups in PcoA using PERMANOVA, with sample distances calculated using the Bray-Curtis method. (B) Analysis of the Microbial Dysbiosis Index (MDI), which evaluates the extent of gut microbial disruption in experimental groups relative to the healthy baseline (CT, control). A two-tailed Wilcoxon rank-sum test was used for inter-group MDI comparison at the OTU level, with P-values adjusted by the false discovery rate (FDR) method. (C) Venn diagram highlighting shared and unique microbial species across the gut microbiota of different experimental groups. (D) Stacked bar chart displaying the relative abundance of gut microbial taxa at the genus level, focusing on the top 15 most abundant genera in each group. (E) Bar chart illustrating species-level variations in gut microbiota composition, emphasizing the top 15 most abundant species across groups. (F–L) Intergroup comparisons of species-level differences in key taxa: *norank_f__Muribaculaceae* (F), *Lactobacillus* (G), *norank_o__Clostridia_UCG-014* (H), *Akkermansia* (I), *Desulfovibrio* (J), *Escherichia_Shigella* (K), and *Sutterella* (L). Data are expressed as means ± SD (*n* = 5). Statistical analysis was performed using one-way ANOVA followed by Tukey's multiple comparison test. Significant differences compared with the DSS group are denoted as *P < 0.05, **P < 0.01, ***P < 0.001, ****P < 0.0001.

two groups (Fig. 4A). Among these, 10 probiotic taxa were specifically enriched in the SupCBAKK group, including: *p__Firmicutes*, *g__Lactobacillus*, *f__Lactobacillaceae*, *o__Lactobacillales*, *g__Desulfovibrio*, *g__Marvinbryantia*, *g__Candidatus_Saccharimonas*, *f__Saccharimonadaceae*, *o__Saccharimonadales*, and *c__Saccharimonadia*. Most of these taxa belong to the core microbiota with intestinal mucosal repair functions (Fig. 4A).

Furthermore, Spearman correlation network analysis revealed that the key microbiota induced by SupCBAKK, such as *Bacteroidales_bacterium_f_Muribaculaceae* and *unclassified_o_Clostr-idia_UCG-014*, were significantly negatively correlated with pro-inflammatory factors IL-1β and IL-6, whereas *unclassified_f_Muribaculaceae* showed a strong positive correlation with the intestinal barrier marker ZO-1 (Fig. 4B). This suggests that SupCBAKK can synergistically regulate key beneficial microbiota, such as *Bacteroidales_bacterium_f_Muribaculaceae* and *unclassified_f_Muribaculaceae* from the *Bacteroidetes* phylum and *unclassified_o_Clostridia_UCG-014* from the *Firmicutes* phylum (which degrade dietary fiber and produce SCFAs, thereby suppressing intestinal inflammation [26, 27]). This establishes a multidimensional "microbiota-immune-barrier" regulatory mechanism. On one hand, it inhibits NF-κB pathway activity to reduce the levels of pro-inflammatory factors IL-6 and IL-1β; on the other hand, it enhances intestinal barrier integrity by promoting the expression of the tight junction protein ZO-1 (Fig. 4B). In summary, these multi-level synergistic microbiota interaction patterns indicate that compared with single-strain metabolite treatment, SupCBAKK can more effectively reshape the gut microbiota of IBD mice through metabolic regulation, immune balance, and intestinal barrier repair, thereby alleviating DSS-induced colitis in mice.

## SupCBAKK combined with aPD-L1 alleviates tumor progression in AOM/DSS-induced CRC mice

IBD often precedes the development of CRC and significantly increases its incidence. This study found that SupCBAKK exhibits remarkable efficacy in alleviating IBD symptoms, suppressing inflammatory factors, and restoring the gut microbiota. Therefore, to further evaluate the role of SupCBAKK in improving colitis-associated CRC progression, we

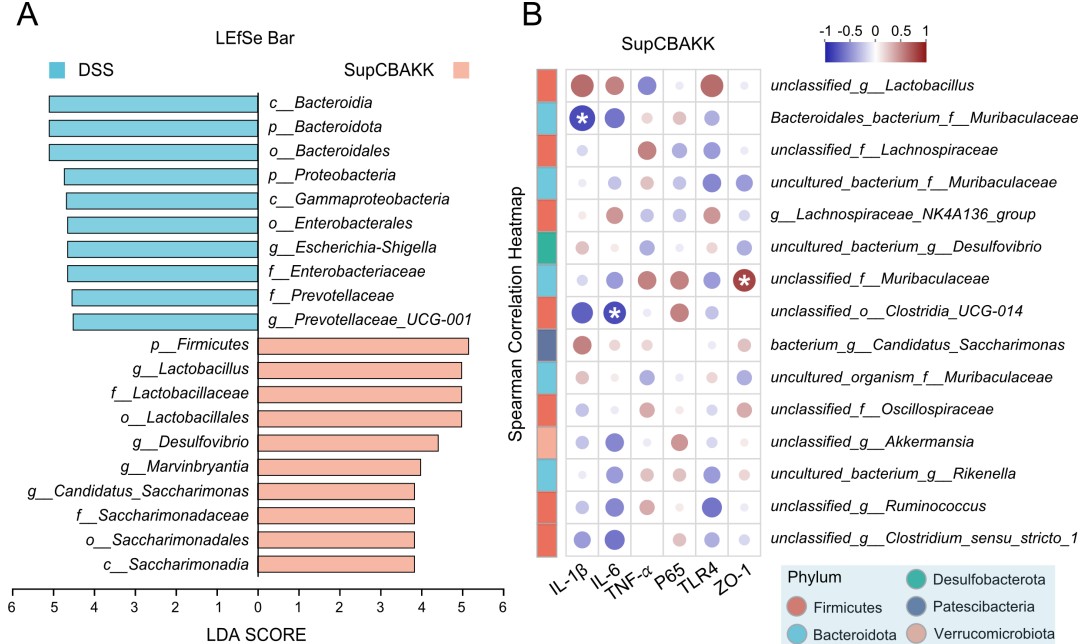

**FIG 4** SupCBAKK modulates the gut microbiota-immune axis to alleviate colitis in mice. (A) Linear discriminant analysis (LDA) histogram illustrating the distribution patterns between DSS and SupCBAKK groups (LDA [log10] > 4, *P* < 0.05). (B) Spearman correlation analysis demonstrating the relationship between gut microbiota composition in SupCBAKK-treated mice and clinical parameters. Data are expressed as mean ± SD (*n* = 5).

established an *in situ* CRC model in mice using the AOM/DSS induction method (Fig. 5A). Additionally, our previous research demonstrated that probiotics combined with the immunotherapeutic agent aPD-L1 had significant synergistic effects in alleviating CRC. Thus, we constructed a combined treatment model of SupCBAKK and aPD-L1 (SupCBAKK + aPD-L1) (Fig. 5A).

We first compared changes in general physiological parameters among the groups, including body weight, colon tumor burden, colon tumor area, colon length, spleen-to-body weight ratio, and survival rate. The results showed that compared with the disease control group, the SupCBAKK treatment group exhibited a 17.52% increase in body weight, 59.86% reduction in colon tumor burden ($P = 0.009$), 55.45% reduction in colon tumor area ($P = 0.0037$), 43.21% increase in colon length ($P < 0.0001$), and 72.07% reduction in spleen-to-body weight ratio ($P = 0.023$) (Fig. 5B through G). Similarly, the combined treatment group (SupCBAKK + aPD-L1) showed significant improvements compared with the disease control group (AOM/DSS): a 36.32% increase in body weight ($P = 0.0013$), 89.41% reduction in colon tumor burden ($P = 0.0002$), 90.31% reduction in colon tumor area ($P = 0.0010$), 36.61% increase in colon length ($P = 0.0006$), and 89.20% reduction in spleen-to-body weight ratio ($P = 0.0175$) (Fig. 5B through G). Notably, the therapeutic efficacy of the combined group was significantly superior to that of the single treatment groups (aPD-L1 or SupCBAKK) in all measured parameters. Furthermore, we compared the impact of different treatment modalities on mouse survival rates; at the end of the experiment, the survival rates of mice in the fermentation supernatant group (SupCBAKK) and the combined group (SupCBAKK+aPD-L1) both increased to 100% (Fig. 5H).

To further assess the effects of SupCBAKK intervention on pathological symptoms in mice, H&E staining was used to analyze the effects of aPD-L1, SupCBAKK, and SupCBAKK + aPD-L1 combined treatment on the restoration of colonic mucosal barrier function. The results showed that the SupCBAKK + aPD-L1 combined group significantly outperformed the single treatment groups in improving colonic mucosal damage, abnormal epithelial cell proliferation, and inflammatory cell infiltration (Fig. 5I). Furthermore, the IHC was employed to analyze the changes in the expression of Ki67 and Caspase 3 proteins in each group. The results showed that Ki67 expression decreased by 29.58%, 51.11% ($P = 0.003$), and 55.36% ($P = 0.0014$) in the aPD-L1, SupCBAKK, and SupCBAKK + aPD-L1 groups, respectively. Similarly, Caspase 3 expression decreased by 41.12% and 49.34% ($P = 0.00422$), and 53.58% ($P = 0.00122$) in the aPD-L1, SupCBAKK, and SupCBAKK +aPD-L1 groups, respectively (Fig. 5J). Excitingly, SupCBAKK fermentation metabolites not only significantly inhibited CRC tumor progression but also enhanced the sensitivity of CRC mice to the immune checkpoint inhibitor aPD-L1, improving its anti-tumor efficacy. This provides a new research direction for the clinical translation of postbiotic-immune combination therapy.

## SupCBAKK enhances the sensitivity of CRC mice to aPD-L1, promoting antitumor immune responses

To further evaluate the impact of SupCBAKK and SupCBAKK + aPD-L1 treatment on the immune response in CRC mice, we first analyzed changes in the composition of peripheral blood cells. The results revealed that, except for Neutrophils, the levels of white blood cells (WBC), granulocytes (GRAN), and Lymphocytes in the peripheral blood of all treatment groups were significantly reduced (Fig. 6A). However, the combination intervention group (SupCBAKK + aPD-L1) exhibited significantly superior outcomes compared with the monotherapy group (Fig. 6A).

Macrophages are the primary immune cells that mediate immune suppression and are capable of inhibiting antitumor immune responses. In inflammation, CD8$^+$ T cells can recruit macrophages, thereby promoting inflammation. Therefore, this study used FCM to detect changes in immune cells in the spleens of mice across different groups. Compared with the disease control group, the number of macrophages in the spleen of treated mice decreased, albeit without significant differences (Fig. 6B). However, the

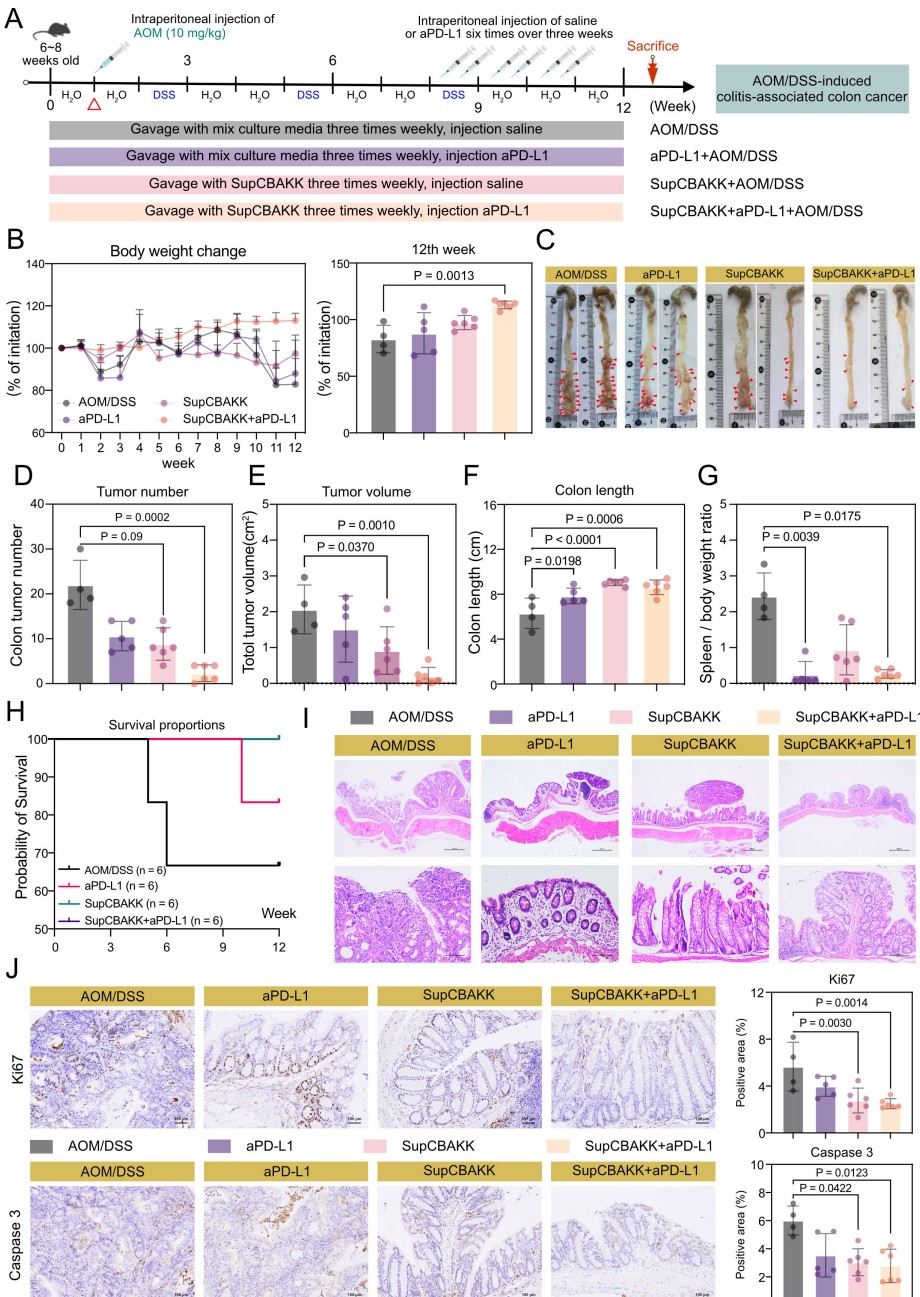

**FIG 5** Combination of SupCBAKK and aPD-L1 significantly suppresses tumor progression in AOM/DSS-induced colitis-associated CRC mice. (A) Experimental schematic of AOM/DSS-induced colitis-associated CRC mouse model establishment and treatment protocol. (B) Temporal changes in body weight of CRC mice. (C) Representative images of colon anatomy from CRC mice. (D) Analysis of tumor number in the colonic tissue. (E) Measurement of total tumor area. (F) Measurement of colon length. (G) Spleen-to-body weight ratio in CRC mice. (H) Survival analysis of CRC mice. (I) Representative H&E-stained histological sections of CRC mice (scale bars: 500 µm and 100 µm). (J) Immunohistochemical analysis of Ki67 and Caspase 3 expression in colonic tissues (scale bar: 100 µm). Data are expressed as means ± SD. Statistical analysis was performed using one-way ANOVA, followed by Tukey's multiple comparison test. Significant differences compared with the AOM/DSS group are denoted as *$P < 0.05$, **$P < 0.01$, ***$P < 0.001$, ****$P < 0.0001$.

proportion of CD8$^+$ T lymphocytes exhibited notable changes: the SupCBAKK intervention group showed a 15.34% reduction ($P = 0.0068$) and the SupCBAKK+aPD-L1 combination intervention group showed a 23.26% reduction ($P = 0.0126$) compared with the disease control group. Moreover, the combination intervention group demonstrated

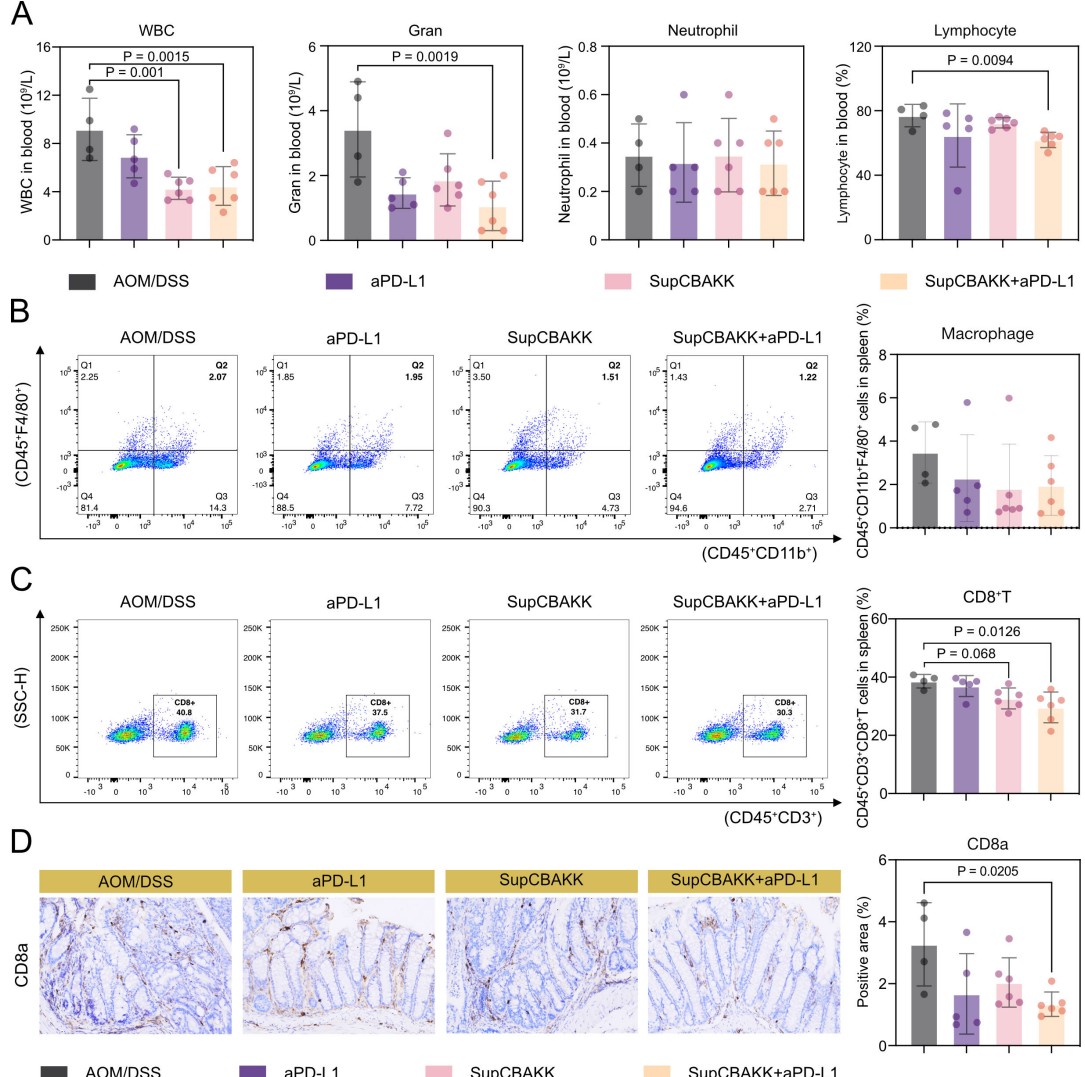

**FIG 6** Synergistic effect of SupCBAKK and aPD-L1 in enhancing anti-tumor immunity in CRC mice. (A) Hematological analysis of peripheral blood parameters, including absolute white blood cell count, granulocyte count, neutrophil count, and lymphocyte percentage. Flow cytometry analysis of (B) macrophage and (C) CD8$^+$ T lymphocyte populations in splenic tissues. (D) Immunohistochemical quantification of CD8a-positive areas in colonic tissues. Data are expressed as means ± SD. Statistical analysis was performed using one-way ANOVA followed by Tukey's multiple comparison test. Significant differences compared with the AOM/DSS group are denoted as *$P < 0.05$, **$P < 0.01$, ***$P < 0.001$, ****$P < 0.0001$.

significantly better efficacy than the monotherapy group (Fig. 6C). Additionally, IHC results confirmed that both the SupCBAKK and SupCBAKK + aPD-L1 groups reduced the inflammatory infiltration of CD8$^+$ T lymphocytes in the colonic tissues of CRC mice, with the combination therapy group exhibiting the most pronounced alleviation (Fig. 6D). In summary, this study indicates that SupCBAKK can reduce the systemic or local intestinal inflammatory infiltration of macrophages and CD8$^+$ T cells, and this effect can be significantly enhanced in synergy with aPD-L1.

## SupCBAKK combined with aPD-L1 modulates gut microbiota composition and increases microbial diversity in CRC mice

Gut microbiota dysbiosis is closely linked to inflammatory responses and tumor progression. To evaluate the impact of combined intervention with SupCBAKK and aPD-L1 on the gut microbiota of CRC mice, we collected fecal samples from both the disease and intervention groups and compared the changes in the gut microbiota using

16S rDNA amplicon sequencing. Through PCoA, Upset analysis, and α-diversity analysis (ACE, Sobs, Chao, and Shannon indices), we found that compared with the disease control group, the gut microbiota diversity in the intervention group was significantly increased (Fig. 7A). Combined treatment with SupCBAKK and aPD-L1 not only altered the gut microbiota composition in CRC mice but also increased the total number of species (Fig. 7B and C).

Notably, both LDA and species composition analysis demonstrated that the combined treatment not only significantly reduced the abnormally elevated levels of *Lactobacillus* in the disease control group but also increased the abundance of beneficial genera such as *Clostridia_UCG-014* and *norank_f__Muribaculaceae* (Fig. 7D through F). Furthermore, Spearman correlation network analysis revealed a strong positive correlation between *s__unclassified_f__Lachnospiraceae* and *s__unclassified_g__Lachnospiraceae_NK4A136_group* in the gut and the proportion of CD8$^+$ T cells in the spleen. In contrast, significantly enriched *s__unclassified_f__Muribaculaceae* showed a strong negative correlation with the proportion of CD8$^+$ T cells in the spleen (Fig. 7G). Additionally, we observed a negative correlation between *s__uncultured_bacterium_g__Dubosiella*, *s__uncultured_bacterium_g__Desulfovibrio*, and *s__unclassified_g__Faecalibaculum* in the gut and the proportion of macrophages in the spleen (Fig. 7G). The metabolites of *s__uncultured_bacterium_g__Dubosiella* can modulate immune responses to alleviate IBD-related symptoms, whereas *s__uncultured_bacterium_g__Desulfovibrio* and *s__unclassified_g__Faecalibaculum* are important producers of butyrate in the gut. These changes in gut microbiota indicate that the combined intervention of SupCBAKK and aPD-L1 not only significantly improves the composition, abundance, and diversity of gut microbiota in CRC mice but also suggests that these structural changes in gut microbiota are closely related to the modulation of immune responses and the suppression of CRC progression.

## DISCUSSION

Our previous studies demonstrated that CB, AKK, and their combination (CBAKK) exhibit significant therapeutic effects in IBD and CRC, with the dual-strain combination being more effective than single strains (9). Based on these findings, this study further evaluated the potential of their fermentation supernatants (SupCB, SupAKK, and SupCBAKK). We found that SupCBAKK markedly alleviated intestinal inflammation, restored barrier function, modulated gut microbiota, and suppressed tumor development. Moreover, SupCBAKK synergized with aPD-L1, sensitizing CRC mice to immune checkpoint blockade and improving anti-tumor efficacy.

A key observation is that SupCBAKK exhibited superior efficacy compared with individual metabolites. Although SupCB and SupAKK partially relieved colitis and reshaped microbiota, SupCBAKK produced stronger effects by reducing pro-inflammatory cytokines (IL-1β, IL-6, and TNF-α), improving barrier integrity (upregulating ZO-1), and suppressing the TLR4/NF-κB pathway (downregulating TLR4 and P65). Since TLR4 activation triggers NF-κB–mediated transcription of inflammatory cytokines, targeted inhibition of this axis appears central to the synergistic mechanism. Given the crosstalk of NF-κB with MAPK and JAK/STAT signaling, SupCBAKK may exert broader immunomodulatory effects. Future studies should determine whether its metabolites directly interfere with TLR4 ligand binding or act on adaptor proteins (e.g., MyD88) and explore possible interactions with complementary inflammatory pathways.

Given the remarkable efficacy of SupCBAKK in treating IBD and CRC, it may be an excellent candidate for the development of next-generation postbiotics (12, 13). During fermentation, CB and AKK release various beneficial components, including short-chain fatty acids (SCFAs, such as butyrate, acetate, and lactate), amino acids, enzymes, and polysaccharides. These metabolites, which are key components of postbiotics, possess multiple bioactive properties and are primarily responsible for alleviating IBD and CRC symptoms (14, 15). For example, butyrate, a major energy source for intestinal epithelial cells, plays a vital role in maintaining intestinal barrier integrity, regulating immune

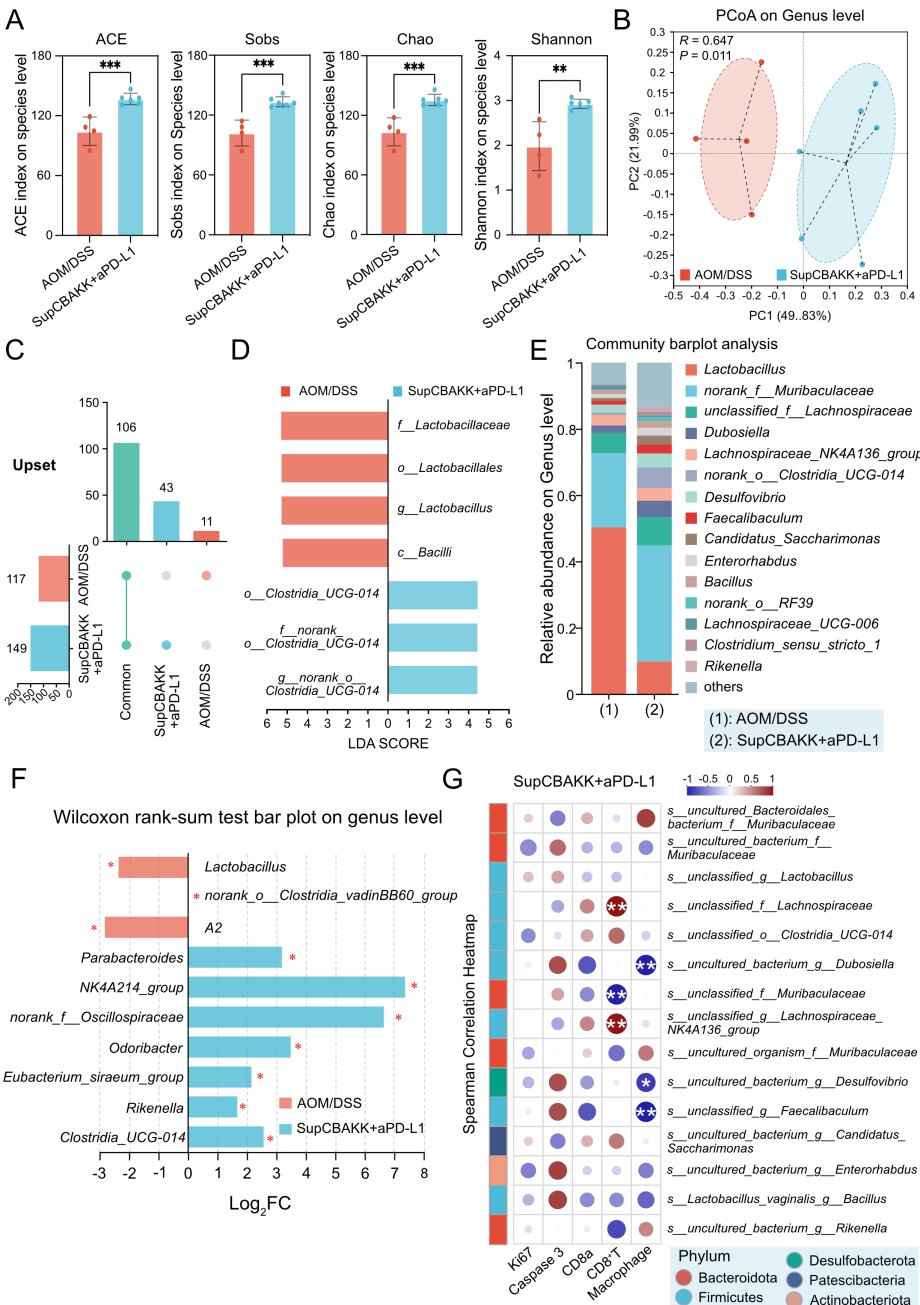

**FIG 7** Impact of SupCBAKK+aPD-L1 combination on gut microbiota composition in AOM/DSS-Induced colitis-associated CRC mice. (A) Comparative α-diversity analysis between AOM/DSS and SupCBAKK + aPD-L1 groups, including ACE, Sobs, Chao, and Shannon indices (Wilcoxon rank-sum test，$P \leq 0.05$ marked as *, $P \leq 0.01$ marked as **, $P \leq 0.001$ marked as ***). (B) Differential assessment of the relative abundance of fecal microbiota among two groups in PcoA using PERMANOVA, with sample distances calculated using the Bray-Curtis method. (C) Upset plot visualization of shared and unique microbial species between groups. (D) LDA histogram showing significantly different taxa (LDA [log10] > 4, $P < 0.05$). (E) Stacked bar plot representing the relative abundance of the top 15 microbial genera. (F) Species-level compositional differences in gut microbiota between groups were compared by the Mann-Whitney U test. (G) Spearman correlation heatmap between gut microbiota profiles and clinical factors in the SupCBAKK + aPD-L1 group.

responses, reducing inflammatory factors, and suppressing tumor progression (15, 28). Acetate, a primary product of dietary fiber fermentation by the gut microbiota, promotes the proliferation and differentiation of intestinal epithelial cells, prevents the invasion

of harmful substances and pathogens, and maintains intestinal homeostasis (29). The Amuc_1100 protein, which is derived from the outer membrane of AKK bacteria, contributes to the regeneration of the intestinal mucus layer, enhances the expression of tight junction proteins in intestinal epithelial cells, promotes the differentiation of regulatory T cells, and improves insulin sensitivity (25, 30, 31). In summary, the numerous beneficial metabolites produced by CB and AKK provide the foundation for the development of novel antibiotics.

Beyond direct effects, postbiotics may remodel the gut ecosystem by favoring beneficial taxa such as *A. muciniphila* and *Lactobacillus*. SCFAs stimulate mucin secretion, reinforce the epithelial barrier, lower luminal pH, and suppress pathogens, thereby expanding ecological niches for commensals (25, 30). In addition, attenuation of NF-κB–driven inflammation and enhanced cross-feeding interactions may further stabilize microbial communities (32). Such host–microbe and microbe–microbe interactions likely underlie the microbiota remodeling observed in this study.

Postbiotics are a class of complex metabolites produced by probiotics, which exhibit beneficial effects. They possess synergistic effects that individual metabolites lack, more closely resemble real physiological conditions, and exhibit high bioactivity and application value. In this study, we combined complex metabolites derived from CB and AKK (SupCB + SupAKK) to formulate an even more complex metabolite mixture (SupCBAKK). We found that this mixture was far more effective in modulating DSS-induced colitis and CRC in mice than single metabolite mixtures, indicating a strong synergistic effect between SupCB and SupAKK. This study provides new insights into the development of novel antibiotics.

Additionally, as the primary functional components of postbiotics, metabolites offer several advantages over live bacteria, including greater stability, ease of storage and transportation, and resistance to variations in temperature and humidity. Moreover, they do not rely on bacterial proliferation and avoid biosafety concerns, making them increasingly attractive for research and applications in food, health products, and pharmaceuticals (33, 34). This study found that SupCBAKK not only replicates the intestinal probiotic activity of live bacteria in treating IBD and CRC but also addresses the stability and safety issues associated with live bacteria. Therefore, postbiotics may be superior to probiotics in clinical applications and translational research.

Despite the promising results, the clinical translation of postbiotics like SupCBAKK faces several challenges that warrant discussion. First, scalable production of complex postbiotic mixtures requires rigorous optimization of fermentation conditions, downstream processing, and standardization of bioactive components to ensure batch-to-batch consistency (35). Second, metabolic stability *in vivo* remains to be fully characterized; the absorption, distribution, and persistence of these metabolites in the human gastrointestinal environment may differ from murine models due to differences in digestive physiology and microbiome composition (36). Third, host microbiome heterogeneity across human populations could lead to variable treatment responses, necessitating personalized formulation or dosing strategies (37). Furthermore, although this study focused on TLR4/NF-κB inhibition, the precise mechanisms underlying the synergy between SupCB and SupAKK require deeper molecular profiling to identify key active components and their targets. Advanced multi-omics approaches (metabolomics and proteomics) and systems biology analyses will be essential to decode the complex mode of action and ensure reproducible efficacy (38). Finally, comprehensive safety assessments, including potential off-target effects or interactions with host metabolism, must be conducted before clinical application (39).

This study provides new research directions for achieving "live bacteria-free" treatment of IBD and CRC and for developing "postbiotic-immune synergy" therapies. It also offers new strategies for the development and application of novel postbiotic products. As research progresses, SupCBAKK may not only play a significant role in treating intestinal-related diseases but also show potential in the prevention and treatment of other conditions, such as metabolic and psychiatric disorders. Addressing

the aforementioned challenges through future mechanistic and translational studies will be crucial to realizing the full clinical potential of postbiotics.

## MATERIALS AND METHODS

### Bacteria and culture condition

The strains used in this study were *Clostridium butyricum* CGMCC0313-1 (CB) and *Akkermansia muciniphila* ATCC BAA-835 (AKK). These strains were anaerobically cultured at 37℃ using Reinforced Clostridial Medium (RCM) and Thioglycollate Medium (TGM), respectively, following standard protocols. The bacterial fermentation supernatants were prepared as follows:

1. Preparation of CB fermentation supernatant (SupCB): A single colony of activated CB was inoculated into 5 mL of antibiotic-free RCM medium and anaerobically cultured at 37℃. When the OD600 reached 0.6 (approximately $1.5 \times 10^8$ CFU/mL), the culture was centrifuged at $5,000 \times g$ for 5 min to collect the supernatant. The CB fermentation supernatant was then filtered through a 0.22 µm sterile filter and stored at −80℃ for future use.
2. Preparation of AKK fermentation supernatant (SupAKK): A single colony of activated AKK was inoculated into 5 mL of antibiotic-free RCM medium and anaerobically cultured at 37℃. When the OD600 reached 0.8 (approximately $1.5 \times 10^9$ CFU/mL), the culture was centrifuged at $5,000 \times g$ for 5 min to collect the supernatant. The AKK fermentation supernatant was then filtered through a 0.22 µm sterile filter and stored at −80℃ for future use.

### Construction of the DSS-induced colitis mouse model

Female C57BL/6 mice aged 6–8 weeks and weighing 18 ± 2 g each (purchased from Hangzhou Ziyuan Laboratory Animal Technology Co., Ltd., Hangzhou, Zhejiang, China) were housed under standard conditions (humidity: 51% ± 13%, temperature: 23℃ ± 3℃, 12/12 h light-dark cycle). After 1 week of acclimatization, the mice were orally administered 100 µL of CB fermentation metabolites (SupCB group), AKK fermentation metabolites (SupAKK group), or a mixture of CB and AKK fermentation metabolites (SupCBAKK group) via gavage every 2 days for 2 weeks (a total of 7 administrations) (40). Mice receiving only the culture medium or physiological saline served as the disease control group (DSS group) and the healthy control group (Health group), respectively. Each group consisted of 6 mice. One week later, except for the healthy control group, all other groups were provided with drinking water containing 2.5% DSS (MW: 36–50 kDa, Cat# 60,316ES60, Yeasen Biotech Co., Ltd, Shanghai, China) to induce a colitis model. The DSS-containing water was administered for 1 week, followed by a return to normal drinking water. On day 9, the mice were euthanized. During the modeling process, the body weight of the mice in each group was recorded daily, and fresh fecal samples were collected to assess occult blood. At the end of the intervention, the distal colon and spleen tissues were collected for subsequent evaluation. All animal experiments conducted in this study were performed in compliance with relevant regulations and were approved by the Animal Care and Use Committee of Guizhou Medical University (Reg. No. of Experimental Facilities Certification: SYXK (Qian) 2018-000).

### Construction of the AOM/DSS-induced colon cancer (colitis-associated CRC) mouse model

Female C57BL/6 mice aged 6–8 weeks and weighing 18 ± 2 g each (purchased from Hangzhou Ziyuan Laboratory Animal Technology Co., Ltd., Hangzhou, Zhejiang, China) were housed under standard conditions (humidity: 51% ± 13%, temperature: 23℃ ± 3℃, 12/12 h light-dark cycle). After 1 week of acclimatization, all mice were intraperitoneally injected with AOM (10 mg/kg) and continued to be fed for 1 week. Subsequently, the

mice were provided with drinking water containing 2.5% (wt/vol) DSS (MW: 36–50 kDa, Cat# 60,316ES60, Yeasen Biotech Co., Ltd., Shanghai, China) for 1 week, followed by a return to normal drinking water for 14 days. This cycle was repeated three times to establish the AOM/DSS-induced CRC model (41).

During the modeling process, the mixed supernatant group (SupCBAKK group) was orally administered 100 µL of a mixture of CB and AKK fermentation metabolites three times per week for a total of 12 weeks. The immunotherapy group (aPD-L1 group) received intraperitoneal injections of 100 µL aPD-L1 (15 mg/kg) twice weekly from week 8 to week 11 (a total of six injections). The combination therapy group (SupCBAKK + aPD-L1 group) received both the mixed supernatant metabolites and aPD-L1. Mice receiving only the mixed culture medium served as the disease control group (AOM/DSS group). Each group consisted of six mice. During the experiment, the severity of colon damage was quantitatively assessed by monitoring changes in body weight, fecal occult blood, and stool consistency. At the end of the intervention, distal colon and spleen tissues were collected for subsequent evaluation, and fecal samples were collected for subsequent metagenomic analysis. All animal experiments conducted in this study were performed in compliance with relevant regulations and were approved by the Animal Care and Use Committee of Guizhou Medical University (Reg. No. of Experimental Facilities Certification: SYXK (Qian) 2018-000).

## Cytokine detection

Colon tissues were collected and homogenized in RIPA tissue lysis buffer according to the manufacturer's instructions (Solarbio Life Science, China, Cat: #R0010, Lot No: 240003004). The homogenates were then centrifuged at 5,000 × $g$ for 5 min at 4℃ to isolate the supernatant. The concentrations of TNF-α, IL-6, and IL-1β cytokines in the colon tissues were measured using Enzyme-Linked Immunosorbent Assay (ELISA) kits (Cat# ml098430, mIC50300-1, mIC50536-1, Shanghai Enzyme Linked Biotechnology Co., Ltd., Shanghai, China).

## Blood cell counts

Mice were fasted for 12 h, and after anesthesia, blood samples were collected from the retro-orbital venous plexus (with anticoagulant treatment). The blood samples were then analyzed using an automated hematology analyzer to count neutrophils, white blood cells (WBC), granulocytes (gran), and lymphocytes.

## Histology and immunohistochemistry (IHC)

Hematoxylin and eosin staining (H&E) and IHC analyses were performed by Wuhan Huayan Biotechnology Co., Ltd. In the IBD model, IHC staining was used to detect the expression of the tight junction protein zonula occludens-1 (ZO-1). In the CRC model, IHC staining was employed to assess the expression of Ki67, Caspase 3, and CD8a proteins. The IHC results were analyzed and processed using ImageJ software (Fiji for Mac OS X, US National Institutes of Health, Bethesda, MD).

## 16S rRNA high-throughput sequencing

Total bacterial DNA was extracted using a Fecal Microbiome DNA Extraction Kit (Lot# Y1904, Tiangen Biotech, Co., Ltd., Beijing, China). The hypervariable V3 region of the 16S rRNA gene was amplified using primers 338F (5′-ACTCCTACGGGAGGCAGCA-3′) and 806R (5′-GGACTACHVGGGTWTCTAAT-3′). Sequencing was performed on the Illumina MiSeq platform (PE300, San Diego, USA) and conducted by Shanghai Majorbio Bio-pharm Technology Co., Ltd. according to standard protocols.

## Western blot

Colon tissues from mice were homogenized in RIPA lysis buffer (Solarbio Life Science, China, Cat: #R0010, Lot No: 240003004) and centrifuged at 12,000 × $g$ for 10 min at 4°C to collect the supernatant. Protein concentrations were quantified using a BCA assay kit (Solarbio Life Science, China, Cat: #PC0020, Lot No: 220191022). Both bacterial and tissue-extracted proteins were separated by 10%–12% SDS-PAGE and transferred to Immobilon®-P PVDF membranes (Merck Millipore Ltd., Tullagreen, Carrigtwohill, Co. Cork, Ireland, Cat: #IPVHO0010, Lot No: BM4AB8360A). The membranes were blocked with 5% non-fat milk in TBST (Tris-buffered saline with 0.1% Tween 20) for 1 h at room temperature. Subsequently, the membranes were incubated with appropriately diluted primary antibodies overnight at 4°C, washed once with TBST, and then incubated with horseradish peroxidase (HRP)-conjugated secondary antibodies diluted in 1% milk-TBST for 1 h at room temperature. The antibodies used included: anti-β-actin (1:2,000; Servicebio, Cat: GB15003-100, Lot No: AC240326070), anti-TLR4 (1:1,000; Proteintech, Cat No: 66350-1-Ig, Lot: 10023998), anti-P65 (1:2,000; Proteintech, Cat No: 80979-1-RR, Lot: 23002541).

## Flow cytometry (FCM)

Spleen tissues were homogenized and passed through a 70-μm sterile cell strainer to obtain single-cell suspensions. The cells were stained with fluorescently labeled antibodies and analyzed using a BD FACSCelesta flow cytometer (Becton, Dickinson, and Company, CA, USA) according to the manufacturer's instructions to quantify the number and proportion of specific cell populations. Data analysis was performed using FlowJo software (version 10, Becton, Dickinson and Company, CA, USA). Antibodies for CD8[+] T Cell Staining: Purified Rat Anti-Mouse CD16/CD32 (Cat# 553141, RRID: AB_2870011, Becton, Dickinson and Company, CA, USA), APC-Cy7 Rat Anti-Mouse CD45 (Cat# 557659, RRID: AB_396774, Becton, Dickinson and Company, CA, USA), FITC Hamster Anti-Mouse CD3ε (Cat# 553061, RRID: AB_394595, Becton, Dickinson and Company, CA, USA), and PerCP-Cy5.5 Rat Anti-Mouse CD8a (Cat# 551162, RRID: AB_394081, Becton, Dickinson and Company, CA, USA). Antibodies for macrophage staining: purified rat anti-mouse CD16/CD32 (Cat# 553141, RRID: AB_2870011, Becton, Dickinson and Company, CA, USA), APC-Cy7 Rat Anti-Mouse CD45 (Cat# 557659, RRID: AB_396774, Becton, Dickinson and Company, CA, USA), FITC Rat Anti-Mouse CD11b (Cat# 553310, RRID: AB_396679, Becton, Dickinson and Company, CA, USA), and APC Rat Anti-Mouse F4/80 (Cat# 566787, RRID: AB_2869866, Becton, Dickinson and Company, CA, USA).

## Data and statistical analysis

All samples and animal experimental data used in this study were included in the analysis. For comparisons between two groups, either Student's $t$-test or non-parametric tests were selected based on the characteristics of the data. When multiple groups were compared, one-way analysis of variance (one-way ANOVA) was first applied, followed by Tukey's multiple comparison test to further clarify intergroup differences. In the selection of statistical analysis methods, if the sample data characteristics satisfied the assumption of normal distribution, one-way ANOVA was preferentially used; otherwise, the Kruskal-Wallis test was employed if the assumption of normal distribution was not met. All results were expressed as mean ± standard deviation, with $P < 0.05$ considered statistically significant. All statistical analyses were performed using GraphPad Prism software (version 9.0, GraphPad Software, San Diego, CA).

## ACKNOWLEDGMENTS

This work was supported by the National Natural Science Foundation of China (32160015), Guizhou Health Commission Project (2025GZWJKJXM0153), High-level Innovation Talent Project of Guizhou (GCC[2023]002; GCC[2022]036-1), Excellent Young Talent Program of Guizhou Medical University (2022)101 and (2022)112, Guizhou Province Key Laboratory (ZDSYS[2023]004), Scientists Workstation Guizhou Province

(KXJZ[2024]009), and The Local Science Foundation of Guizhou Province Guided by Central Committee of China (Qiankehe [2025]024).

G.C., Z.C., and W.H. were involved in conceptualization, revised the manuscript, and supervised the findings of this work. D.H., Q.Y., and X.Z. conceived and designed the analysis. D.H. and D.W. performed data curation and formal analysis. Y.K., L.T., B.L., X.W., and Z.Z. verified the analytical methods. The original draft was written by D.H. and Q.Y. All authors approved the final manuscript. D.H., Q.Y., X.Z., and D.W. contributed equally to this work.

The authors declare that the research was conducted without any commercial or financial relationships that could be construed as a potential conflict of interest.

## AUTHOR AFFILIATIONS

[1]Guizhou Key Laboratory of Microbio and Infectious Disease Prevention & Control, Guizhou Medical University, Guiyang, China

[2]Key Laboratory of Endemic and Ethnic Diseases, Ministry of Education & School/Hospital of Stomatology Guizhou Medical University, Guiyang, China

[3]Collaborative Innovation Center for Prevention and Control of Endemic and Ethnic Regional Diseases Co-constructed by the Province and Ministry, Guiyang, China

[4]Guizhou Provincial Engineering Technology Research Center for Chemical Drug R&D, Guizhou Medical University, Guiyang, China

[5]Collaborative Innovation Center for Prevention and Control of Endemic and Ethnic Regional Diseases Co-constructed by the Province and Ministry of China, Guiyang, China

[6]School of Public Health/Key Laboratory of Endemic and Ethnic Diseases, Ministry of Education &Key Laboratory of Medical Molecular Biology of Guizhou Province, Guizhou Medical University, Guiyang, China

## AUTHOR ORCIDs

Wei Hong http://orcid.org/0000-0002-3317-9401
Zhenghong Chen http://orcid.org/0000-0002-2944-6001
Guzhen Cui http://orcid.org/0000-0002-6520-1052

## FUNDING

| Funder | Grant(s) | Author(s) |
| --- | --- | --- |
| National Natural Science Foundation of China | 32460046 | Guzhen Cui |
| Guizhou Health Commission Project | 2025GZWJKJXM0153 | Guzhen Cui |
| High-level Innovation Talent Project of Guizhou | GCC[2023]002,GCC[2022]036-1 | Yingqian Kang |
| Excellent Young Talent Program of Guizhou Medical University | (2022)101,(2022)112 | Wei Hong |
| Guizhou Provence Key Laboratory | ZDSYS[2023]004 | Zhenghong Chen |
| Scientists Workstation Guizhou Provence | KXJZ[2024]009 | Yingqian Kang |
| The Local Science Foundation of Guizhou Provence Guided by Central Committee of China | Qiankehe [2025]024 | Zhenghong Chen |

## AUTHOR CONTRIBUTIONS

Dengxiong Hua, Data curation, Formal analysis, Investigation, Software, Writing – original draft | Qin Yang, Data curation, Writing – original draft | Xuexue Zhou, Formal analysis, Investigation | Daoyan Wu, Data curation, Formal analysis | Yingqian Kang, Validation | Lei Tang, Validation | Boyan Li, Validation | Zhengrong Zhang, Validation | Xinxin Wang,

Investigation | Wei Hong, Conceptualization, Funding acquisition, Writing – review and editing | Zhenghong Chen, Conceptualization, Funding acquisition, Writing – review and editing | Guzhen Cui, Conceptualization, Funding acquisition, Methodology, Supervision, Writing – review and editing

## DATA AVAILABILITY

All sequencing data were uploaded to the NCBI Sequence Read Archive (https://www.ncbi.nlm.nih.gov/bioproject/1393054) with the accession number BioProject ID PRJNA1393054.

## ADDITIONAL FILES

The following material is available online.

### Supplemental Material

**Supplemental figures (mSystems00689-25-s0001.docx).** Original Western blot images.

### Open Peer Review

**PEER REVIEW HISTORY (review-history.pdf).** An accounting of the reviewer comments and feedback.

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
