## [Reviewer comments · mSystems]

Synergistic effects of *Clostridium butyricum* and *Akkermansia muciniphila*-derived postbiotics ameliorate DSS-induced colitis and associated tumorigenesis through immunomodulation and microbiota regulation in mice

Dengxiong Hua, Qin Yang, Xuexue Zhou, Daoyan Wu, Yingqian Kang, Lei Tang, Boyan Li, Zhengrong Zhang, Xinxin Wang, Wei Hong, Zhenghong Chen, and Guzhen Cui

Corresponding Author(s): Guzhen Cui, Guizhou Medical University

Review Timeline:

Submission Date:	May 13, 2025
Editorial Decision:	September 3, 2025
Revision Received:	September 25, 2025
Accepted:	December 3, 2025

Editor: Emily Cope

Reviewer(s): Disclosure of reviewer identity is with reference to reviewer comments included in decision letter(s). The following individuals involved in review of your submission have agreed to reveal their identity: Yan Wang (Reviewer #1); Bidisha Barat (Reviewer #2)

Transaction Report:

DOI: <https://doi.org/10.1128/msystems.00689-25>

Re: mSystems00689-25 (**Synergistic effects of *Clostridium butyricum* and *Akkermansia muciniphila*-derived postbiotics ameliorate DSS-induced colitis and associated tumorigenesis through immunomodulation and microbiota regulation in mice**)

Dear Dr. Guzhen Cui:

Revision Guidelines

Sincerely,
Emily Cope
Editor
mSystems

Reviewer #1 (Comments for the Author):

The manuscript presents a comprehensive study on the synergistic effects of postbiotics derived from *Clostridium butyricum* and *Akkermansia muciniphila* in ameliorating DSS-induced colitis and colitis-associated tumorigenesis in mice. The research is well-designed, with robust experimental methodologies and clear results. The findings highlight the potential of postbiotics as a

therapeutic alternative to live probiotics, particularly in modulating gut microbiota and immune responses. However, several areas require clarification or improvement to strengthen the manuscript's impact and readability.

Reviewer #2 (Comments for the Author):

This is a well-conceived and executed study investigating the therapeutic potential of *Clostridium butyricum* and *Akkermansia muciniphila* derived postbiotics in models of colitis and colitis-associated colorectal cancer. The authors convincingly demonstrate that combined fermentation products (SupCBAKK) ameliorate DSS-induced colitis and enhance the response to anti-PD-L1 immunotherapy, highlighting a novel postbiotic-immunotherapy synergy. The work is timely, relevant, and addresses an important translational gap between live probiotics and more stable, safer metabolite-based interventions. See attached PDF for specific comments.

The manuscript presents compelling data and has high translational significance, but it would benefit from the following revisions:

Revisions required:

Stronger mechanistic insights into metabolite-level drivers of efficacy.

1. While the data strongly support efficacy, the molecular mechanisms underlying SupCBAKK action remain somewhat superficial. The manuscript mainly shows correlations (e.g., SCFA-producing taxa enrichment, cytokine reduction). Further mechanistic assays (e.g., direct SCFA quantification, metabolomic profiling of the supernatants) would substantially strengthen the conclusions.
2. The role of specific metabolites (butyrate, acetate, Amuc_1100 protein) is discussed but not experimentally confirmed. Even targeted metabolite quantification would add value.

Synergistic effects of *Clostridium butyricum* and *Akkermansia muciniphila*-derived postbiotics ameliorate DSS-induced colitis and associated tumorigenesis through immunomodulation and microbiota regulation in mice

The manuscript presents a comprehensive study on the synergistic effects of postbiotics derived from *Clostridium butyricum* and *Akkermansia muciniphila* in ameliorating DSS-induced colitis and colitis-associated tumorigenesis in mice. The research is well-designed, with robust experimental methodologies and clear results. The findings highlight the potential of postbiotics as a therapeutic alternative to live probiotics, particularly in modulating gut microbiota and immune responses. However, several areas require clarification or improvement to strengthen the manuscript's impact and readability.

1. Reduce the number of keywords to five.
2. Emphasize more clearly in the introduction how SupCBAKK differs from previously reported postbiotics or synbiotics and elaborate on what specific “postbiotic” components are hypothesized to confer the observed synergistic effects. Clarify whether SupCBAKK includes extracellular vesicles, secreted proteins, or small molecules.
3. The results section contains elements of methods, discussion, and references, but it should only present the results.
4. Since several statistical analysis methods were used in this study, please specify in each figure legend which results correspond to which statistical analysis method.
5. While the study shows reduced pro-inflammatory markers and improved microbiota composition, the mechanisms underlying the synergy between SupCB and SupAKK remain largely descriptive. The role of TLR4/NF- κ B suppression in the observed anti-inflammatory effects should be discussed in greater depth, including potential cross-talk with other signaling pathways.
6. The manuscript mentions increased *Akkermansia* and *Lactobacillus* but does not explain how SupCBAKK specifically enriches these taxa. Are these effects direct (e.g., substrate utilization) or indirect (e.g., immune-mediated)?
7. The discussion emphasizes translational potential but does not address key challenges for postbiotic development (e.g., scalability, metabolite stability *in vivo*, human microbiota differences). A paragraph on these limitations would balance the optimism.

Response to Reviewer Comments

Reviewer #1 (Comments for the Author):

The manuscript presents a comprehensive study on the synergistic effects of postbiotics derived from *Clostridium butyricum* and *Akkermansia muciniphila* in ameliorating DSS-induced colitis and colitis-associated tumorigenesis in mice. The research is well-designed, with robust experimental methodologies and clear results. The findings highlight the potential of postbiotics as a therapeutic alternative to live probiotics, particularly in modulating gut microbiota and immune responses. However, several areas require clarification or improvement to strengthen the manuscript's impact and readability.

1. Reduce the number of keywords to five.

Response:

Based on your suggestion and after careful consideration, we have removed the two keywords "antitumor immune". The deletion was made at lines 48-49 in the revised manuscript.

2. Emphasize more clearly in the introduction how SupCBAKK differs from previously reported postbiotics or synbiotics and elaborate on what specific "postbiotic" components are hypothesized to confer the observed synergistic effects. Clarify whether SupCBAKK includes extracellular vesicles, secreted proteins, or small molecules.

Response:

We sincerely thank the reviewer for this very valuable comment. SupCBAKK is substantially different from previously reported postbiotics or synbiotics. The key distinctions are outlined below:

(1) While previous studies have focused on the functional roles of metabolic products from single bacteria (either *Clostridium butyricum* (CB) or *Akkermansia muciniphila* (AKK)) in disease treatment, there have been no reports on the use of dual-bacterial (CB and AKK) mixed fermentation to prepare a composite postbiotic.

(2) Previous functional studies on CB metabolites have primarily emphasized
short-chain fatty acids (e.g., acetate, propionate), which are known to significantly
inhibit pathogen activity, invasiveness, and biofilm formation, as well as modulate
intestinal inflammation and microbial balance(1-4). AKK fermentation metabolites are
also rich in organic acids and lipid metabolites, which hold considerable potential for
pathogen control and metabolic disease intervention(5, 6). Moreover, AKK produces a
unique key outer membrane protein, Amuc_1100, which plays important roles in
enhancing tight junctions, promoting epithelial repair, and alleviating DSS-induced
barrier damage and endotoxemia(7-9). Studies have found that both live and
pasteurized AKK cells exhibit similar efficacy in treating inflammatory bowel disease
(IBD)(10, 11). Thus, both bacterial cells and their metabolites play significant roles in
regulating host immunity and microecological balance.

(3) The main innovation of our study lies in combining the features of two
probiotics, CB and AKK, by mixing their fermented metabolites (termed SupCBAKK).
We discovered that this mixed metabolite preparation (which we refer to as a postbiotic)
is significantly superior to single-bacterium derivatives in alleviating DSS-induced
colitis and colorectal cancer progression. Furthermore, our prior research demonstrated
that a combination of live CB and AKK also yields markedly better outcomes than
single-strain treatments, providing a solid foundation for the current study.

(4) Individual fermentation products of CB and AKK generate various bioactive
components, including extracellular vesicles, secreted proteins (e.g., Amuc_1100 from
AKK), and key small molecules (e.g., butyrate, acetate, lactate)(1, 3, 8, 9). The
combined fermentative product SupCBAKK used in this study also contains these
multiple bioactive components. However, research on postbiotics primarily emphasizes
the holistic function of composite products, which are often more effective than single
components. Therefore, our study does not specifically highlight the efficacy of any
single metabolite.

We have supplemented and clarified the relevant content in both the Introduction
and Discussion sections of the revised manuscript, with all changes highlighted using
the “Track Changes” feature.

**3. The results section contains elements of methods, discussion, and references,**
**but it should only present the results.**

**Response:**

Thank you very much for your professional suggestions. In the revised version, we
have removed the content related to "methodology, discussion, and references" from
the original results section, and have made corresponding additions or adjustments in
the appropriate sections.

**4. Since several statistical analysis methods were used in this study, please**
**specify in each figure legend which results correspond to which statistical analysis**
**method.**

**Response:**

Thank you for your valuable feedback. In response to the analysis methods and
specific requirements of different datasets, we employed distinct statistical approaches
in various results sections, which may have caused confusion for you and the readers.
We have now added the corresponding statistical methods in the relevant results
sections and figure captions of the revised manuscript.

**5. While the study shows reduced pro-inflammatory markers and improved**
**microbiota composition, the mechanisms underlying the synergy between SupCB**
**and SupAKK remain largely descriptive. The role of TLR4/NF- κ B suppression in**
**the observed anti-inflammatory effects should be discussed in greater depth,**
**including potential cross-talk with other signaling pathways.**

**Response:**

Thank you for this insightful suggestion. According to the literature, the primary
functional components in the metabolites of CB and AKK that contribute to human
health include the organic acids, proteins, and other potentially unidentified small
molecule metabolites(12-14). Current research indicates that these metabolic
components are primarily involved in the TLR4/NF- κ B signaling pathway(15-19),
which is why we selected this pathway for investigation.

As you rightly pointed out, whether there is crosstalk with other signaling pathways
remains uncertain based on our current data—such interaction may or may not exist.

Addressing this question would require extensive additional research and data. That
said, in an ongoing study we are conducting on bacterial vesicles (currently in progress),
the application of multi-omics and big data analytics may provide further insights into
this issue.

We greatly appreciate your valuable comment. In response, we have added a
discussion in the revised manuscript elaborating on the role of TLR4/NF- κ B inhibition
in the anti-inflammatory effects, including potential crosstalk with other signaling
pathways. We believe that a more definitive answer may emerge upon the completion
of our additional study.

**6. The manuscript mentions increased *Akkermansia* and *Lactobacillus* but**
**does not explain how SupCBAKK specifically enriches these taxa. Are these effects**
**direct (e.g., substrate utilization) or indirect (e.g., immune-mediated)?**

**Response:**

Thank you for your comment. This study found that although live probiotics were
not directly administered, the supplementation with the probiotic metabolite
SupCBAKK significantly increased the relative abundance
of *Akkermansia* and *Lactobacillus* in the mouse intestine. This enhancement is
considered a key mechanism through which probiotics, prebiotics, or postbiotics
improve host health(20, 21). Beyond direct effects, postbiotics may remodel the gut
ecosystem by favoring beneficial taxa such as *A. muciniphila* and *Lactobacillus*. SCFAs
stimulate mucin secretion, reinforce the epithelial barrier, lower luminal pH, and
suppress pathogens, thereby expanding ecological niches for commensals(22-24). In
addition, attenuation of NF- κ B-driven inflammation and enhanced cross-feeding
interactions may further stabilize microbial communities(25). Such host-microbe and
microbe-microbe interactions likely underlie the microbiota remodeling observed in
this study.

In response to your suggestion, we have expanded the Discussion section to include
an elaboration on the potential relationships among TLR4/NF- κ B signaling, short-chain
fatty acids (SCFAs), and the observed increases in *A. muciniphila* and *Lactobacillus*.
For details, please refer to the revised manuscript.

**7. The discussion emphasizes translational potential but does not address key**
**challenges for postbiotic development (e.g., scalability, metabolite stability *in vivo*,**
**human microbiota differences). A paragraph on these limitations would balance**
**the optimism.**

**Response:**

We fully agree that the Discussion section should provide a more comprehensive
evaluation of the practical challenges and limitations associated with the development
of postbiotics. In accordance with your suggestion, we have added a paragraph to the
Discussion section highlighting key issues such as large-scale production processes,
metabolic stability *in vivo*, and individual variations in the human microbiome, in order
to present a more objective and balanced argument. Your comments have significantly
enhanced the rigor and completeness of our study. We sincerely thank you once again.

**Reviewer #2 (Comments for the Author):**

This is a well-conceived and executed study investigating the therapeutic potential
of *Clostridium butyricum* and *Akkermansia muciniphila* derived postbiotics in models
of colitis and colitis-associated colorectal cancer. The authors convincingly
demonstrate that combined fermentation products (SupCBAKK) ameliorate DSS-
induced colitis and enhance the response to anti-PD-L1 immunotherapy, highlighting a
novel postbiotic-immunotherapy synergy. The work is timely, relevant, and addresses
an important translational gap between live probiotics and more stable, safer
metabolite-based interventions. See attached PDF for specific comments.

The manuscript presents compelling data and has high translational significance,
but it would benefit from the following revisions:

Revisions required:

Stronger mechanistic insights into metabolite-level drivers of efficacy.

**1. While the data strongly support efficacy, the molecular mechanisms**
**underlying SupCBAKK action remain somewhat superficial. The manuscript**
**mainly shows correlations (e.g., SCFA-producing taxa enrichment, cytokine**
**reduction). Further mechanistic assays (e.g., direct SCFA quantification,**
**metabolomic profiling of the supernatants) would substantially strengthen the**
**conclusions.**

**Response:**

We sincerely thank you for your insightful and thorough comments on our study.
In this study, we were surprised to find that SupCBAKK significantly promoted the
enrichment of *Akkermansia* and *Lactobacillus* in the intestines of mice with
inflammation, accompanied by an increase in short-chain fatty acid (SCFA)-producing
bacteria and a decrease in cytokine levels. This finding presents a very interesting and
highly worthwhile topic for in-depth analysis.

Therefore, to further elucidate the role of SCFA-producing bacteria in IBD-related
diseases, we specifically analyzed the effect of supplementing with *Lactobacillus* in
treating mouse colitis in another study, and also examined its impact on SCFA
metabolism (that study used *Lactobacillus casei*, as shown in Figure 1). Furthermore,

since the surface protein Amuc_1100 produced by AKK has effects similar to those of
AKK itself (8), we also designed experiments in another study to investigate the
combined effect of *Lactobacillus* and the Amuc_1100 protein in treating IBD (*L. c-*
*Amuc1100*) , along with an analysis of its impact on SCFA production (this design also
addresses the reviewer's second question). Preliminary results indicate
that *Lactobacillus* alone significantly promotes the biosynthesis of SCFAs in the colon,
particularly leading to a marked increase in butyrate levels. Moreover, with the
synergistic effect of the Amuc_1100 protein, SCFA levels (especially acetate) could be
further elevated (as illustrated in the figure below).

In summary, these results preliminarily suggest that SupCBAKK may exert its anti-
inflammatory effects by promoting the colonization of probiotics (such
as *Lactobacillus* and *Akkermansia*), thereby enhancing the production of the microbial
metabolite SCFAs (as illustrated in the figure below). Therefore, we believe that these
follow-up experiments provide more direct molecular evidence for understanding the
mechanism of action of SupCBAKK.

(Note: Since this data falls within the scope of another research project and is
currently incomplete, we have not included it in the present study. We are confident
that subsequent in-depth research will provide a better and more comprehensive answer
to your question.)

**Figure 1** (A) Bar chart of SCFA levels in the intestines of mice from each group. Comparison of
 individual SCFA concentrations: (B) butanoic acid, (C) acetic acid, (D) valeric acid, (E) propanoic
 acid, (F) isobutyric acid, (G) isohexanoic acid, (H) hexanoic acid, and (I) isovaleric acid.

**2. The role of specific metabolites (butyrate, acetate, Amuc_1100 protein) is**
**discussed but not experimentally confirmed. Even targeted metabolite**
**quantification would add value.**

**Response:**

This point is similar to the first question. To validate this hypothesis, we designed
another study investigating the therapeutic effects of the gut
commensal *Lactobacillus (L.c)*, the Amuc_1100 protein, and their combination (*L.c* +
Amuc1100) on IBD-related conditions. Preliminary experimental results indicate that
administration of either *L.c* or the Amuc_1100 protein alone significantly alleviated
intestinal inflammation and enhanced barrier integrity in mice. Furthermore, the
combined intervention group exhibited a more pronounced synergistic therapeutic
effect, outperforming either single treatment group (as shown in the figure below).
These results preliminarily support our hypothesis that SupCBAKK may exert its
immunomodulatory and barrier-protective effects, at least in part, by promoting the
accumulation of beneficial bacteria and their active metabolites.

(Note: Since the aforementioned data belong to another ongoing study, they are not
presented in detail here. Furthermore, as results from that study are still preliminary and
not yet fully finalized, the data presented here represent interim findings and are not
conclusive.)

**Figure 2** (A) Schematic diagram of DSS-induced colitis modeling and intervention in mice; (B)
 percentage change in body weight during modeling; (C) disease activity index (DAI) scores; (D)
 spleen index; (E) colon length measurement; (G) representative H&E-stained sections and
 histopathological scores. Scale bar: 100 μ m.

**REFERENCES:**

- 1. Stoeva MK, Garcia-So J, Justice N, Myers J, Tyagi S, Nemchek M, McMurdie PJ,
Kolterman O, Eid J. 2021. Butyrate-producing human gut symbiont, *Clostridium butyricum*,
and its role in health and disease. *Gut microbes* 13:1907272.
- 2. Wang LY, He LH, Xu LJ, Li SB. 2024. Short-chain fatty acids: bridges between diet, gut
microbiota, and health. *Journal of Gastroenterology and Hepatology* 39:1728-1736.
- 3. Liu L, Zheng C, Xu Z, Wang Z, Zhong Y, He Z, Zhang W, Zhang Y. 2024. Intranasal
administration of *Clostridium butyricum* and its derived extracellular vesicles alleviate
LPS-induced acute lung injury. *Microbiology Spectrum* 12:e0210824.
- 4. Ma L, Lyu W, Song Y, Chen K, Lv L, Yang H, Wang W, Xiao Y. 2023. Anti-Inflammatory
Effect of *Clostridium butyricum*-Derived Extracellular Vesicles in Ulcerative Colitis:
Impact on Host microRNAs Expressions and Gut Microbiome Profiles. *Molecular*
*Nutrition & Food Research* 67:e2200884.
- 5. Liu C, Ma R, Li H, Pan X, Qian H, Yang T, Tian Y. 2025. *Akkermansia muciniphila*
ameliorates fatty liver through microbiota-derived α -ketoisovaleric acid metabolism and
hepatic PI3K/Akt signaling. *iScience* 28.
- 6. Si J, Kang H, You HJ, Ko G. 2022. Revisiting the role of *Akkermansia muciniphila* as a
therapeutic bacterium. *Gut microbes* 14:2078619.
- 7. Plovier H, Everard A, Druart C, Depommier C, Van Hul M, Geurts L, Chilloux J, Ottman
234 N, Duparc T, Lichtenstein L, Myridakis A, Delzenne NM, Klievink J, Bhattacharjee A, van
der Ark KCH, Aalvink S, Martinez LO, Dumas M-E, Maiter D, Loumaye A, Hermans MP,
Thissen J-P, Belzer C, de Vos WM, Cani PD. 2016. A purified membrane protein from
*Akkermansia muciniphila* or the pasteurized bacterium improves metabolism in obese and
diabetic mice. *Nature Medicine* 23:107-113.
- 8. Wang L, Tang L, Feng Y, Zhao S, Han M, Zhang C, Yuan G, Zhu J, Cao S, Wu Q, Li L,
Zhang Z. 2020. A purified membrane protein from *Akkermansia muciniphila* or the
pasteurised bacterium blunts colitis associated tumourigenesis by modulation of CD8⁺ T
cells in mice. *Gut* 69:1988-1997.
- 9. Wang J, Xu W, Wang R, Cheng R, Tang Z, Zhang M. 2021. The outer membrane protein
Amuc_1100 of *Akkermansia muciniphila* promotes intestinal 5-HT biosynthesis and
extracellular availability through TLR2 signalling. *Food & Function* 12:3597-3610.
- 10. Zhang Q, Peng L, Zhang Q, Guo J, Yu N, Yang J, Zuo W. 2025. Oral *Akkermansia*
*muciniphila* biomimetic nanotherapeutics for ulcerative colitis targeted treatment by
repairing intestinal epithelial barrier and restoring redox homeostasis. *ACS Applied*
*Materials & Interfaces* 17:5942-5954.
- 11. Ding Y, Hou Y, Lao X. 2025. The Role of *Akkermansia muciniphila* in Disease Regulation.
*Probiotics and Antimicrobial Proteins*:1-12.
- 12. Ioannou A, Berkhout MD, Geerlings SY, Belzer C. 2025. *Akkermansia muciniphila*:
biology, microbial ecology, host interactions and therapeutic potential. *Nature Reviews*
*Microbiology* 23:162-177.

- 13. Aja E, Zeng A, Gray W, Connelley K, Chaganti A, Jacobs JP. 2025. Health effects and
therapeutic potential of the gut microbe *Akkermansia muciniphila*. *Nutrients* 17:562.
- 14. Tang X, Zeng Y, Li M. 2025. *Clostridium butyricum*: a promising approach to enhancing
intestinal health in poultry. *Frontiers in Veterinary Science* 12:1544519.
- 15. Xie Q, Liu J, Yu P, Qiu T, Jiang S, Yu R. 2025. Unlocking the power of probiotics,
postbiotics: targeting apoptosis for the treatment and prevention of digestive diseases.
*Frontiers in Nutrition* 12:1570268.
- 16. Mun C, Cai J, Hu X, Zhang W, Zhang N, Cao Y. 2022. *Clostridium butyricum* and Its
Culture Supernatant Alleviate the *Escherichia coli*-Induced Endometritis in Mice. *Animals* :
an Open Access Journal From MDPI 12.
- 17. Wang K, Wang K, Wang J, Yu F, Ye C. 2022. Protective Effect of *Clostridium butyricum*
on *Escherichia coli*-Induced Endometritis in Mice via Ameliorating Endometrial Barrier
and Inhibiting Inflammatory Response. *Microbiology Spectrum* 10:e0328622.
- 18. Zheng B, Wen H, Zheng B, Xiang X, Zhu C. 2025. Dietary Carbohydrates and the Intestinal
Barrier: Emerging Insights into NF- κ B Modulation and Health Outcomes. *Journal of*
*Agricultural and Food Chemistry*.
- 19. Doke R, Chande K, Dingare S, Vinchurkar K, Singh S. 2025. Demystifying the role of
postbiotics in inflammation mediated metabolic disorders: an updated review: R. Doke et
al. *Food Science and Biotechnology*:1-22.
- 20. Yue T, Lu Y, Ding W, Xu B, Zhang C, Li L, Jian F, Huang S. 2025. The Role of Probiotics,
Prebiotics, Synbiotics, and Postbiotics in Livestock and Poultry Gut Health: A Review.
*Metabolites* 15:478.
- 21. Saedi S, Derakhshan S, Hasani A, Khoshbaten M, Poortahmasebi V, Milani PG, Sadeghi
278 J. 2025. Recent advances in gut microbiome modulation: effect of probiotics, prebiotics,
synbiotics, and postbiotics in inflammatory bowel disease prevention and treatment.
*Current Microbiology* 82:12.
- 22. Okumura R, Takeda K. The role of the mucosal barrier system in maintaining gut symbiosis
to prevent intestinal inflammation, p 2. *In* (ed), Springer,
- 23. Sankarganesh P, Bhunia A, Kumar AG, Babu S, Gopukumar S, Lokesh E. 2025. Short-
chain fatty acids (SCFAs) in gut health: Implications for drug metabolism and therapeutics.
*Medicine in Microecology*:100139.
- 24. Mukhopadhyaya I, Louis P. 2025. Gut microbiota-derived short-chain fatty acids and their
role in human health and disease. *Nature Reviews Microbiology*:1-17.
- 25. Chen S, Hu T, Xu L, Li J, Liu C, Lin Q, Cao Z. 2025. *Lactobacillus reuteri* SBC5-3
suppresses TNF- α -induced inflammatory responses via NF- κ B pathway inhibition in
intestinal epithelial cells. *Frontiers in Microbiology* 16:1573479.

Re: mSystems00689-25R1 (**Synergistic effects of *Clostridium butyricum* and *Akkermansia muciniphila*-derived postbiotics ameliorate DSS-induced colitis and associated tumorigenesis through immunomodulation and microbiota regulation in mice**)

Dear Dr. Guzhen Cui:

Your manuscript has been accepted, and I am forwarding it to the ASM production staff for publication. Your paper will first be checked to make sure all elements meet the technical requirements. ASM staff will contact you if anything needs to be revised before copyediting and production can begin. Otherwise, you will be notified when your proofs are ready to be viewed.

Sincerely,
Emily Cope
Editor
mSystems